# [Re] Can gradient clipping mitigate label noise?

**David Mizrahi**
EPFL
david.mizrahi@epfl.ch

**Oğuz Kaan Yüksel**
EPFL
oguz.yuksel@epfl.ch

**Aiday Marlen Kyzy**
EPFL
aiday.marlenkyzy@epfl.ch

## Reproducibility Summary

**Scope of Reproducibility**

The original paper proposes partially Huberised losses, which possess label noise robustness. The authors claim that there exist label noise scenarios that defeat Huberised but not partially Huberised losses, and that partially Huberised versions of existing losses perform well on real-world datasets subject to symmetric label noise.

**Methodology**

All the experiments described in the paper were fully re-implemented using NumPy, SciPy and PyTorch. The experiments on synthetic data were run on a CPU, while the deep learning experiments were run using a Nvidia RTX 2080 Ti GPU. Running the experimentation necessary to gain some insight on some of the network architectures used and reproducing the real-world experiments required over 550 GPU hours.

**Results**

Overall, our results mostly support the claims of the original paper. For the synthetic experiments, our results differ when using the exact values described in the paper, although they still support the main claim. After slightly modifying some of the experiment settings, our reproduced figures are nearly identical to the figures from the original paper. For the deep learning experiments, our results differ, with some of the baselines reaching a much higher accuracy on MNIST, CIFAR-10 and CIFAR-100. Nonetheless, with the help of an additional experiment, our results support the authors' claim that partially Huberised losses perform well on real-world datasets subject to label noise.

**What was easy**

The original paper is well written and insightful, which made it fairly easy to implement the partially Huberised version of standard losses based on the information given. In addition, recreating the synthetic datasets used in two of the original paper's experiments was relatively straightforward.

**What was difficult**

Even though the authors were very detailed in their feedback, finding the exact hyperparameters used in the real-world experiments required many iterations of inquiry and experimentation. In addition, the CIFAR-10 and CIFAR-100 experiments can be difficult to reproduce due to the high number of experiment configurations, resulting in many training runs and a relatively high computational cost of over 550 GPU hours.

**Communication with original authors**

We contacted the authors on multiple occasions regarding some of the hyperparameters used in their experiments, to which they promptly replied with very detailed explanations.

ML Reproducibility Challenge 2020.

# 1 Introduction

Gradient clipping is a well-established technique in machine learning, usually motivated by its benefits in optimization. For example, clipping is used extensively to remedy the well-known problem of exploding gradients (Bengio et al., 1994), commonly faced when training recurrent neural networks. Intuitively, it ensures that the norm of the gradient behaves well under iterates of optimization. Indeed, Zhang et al. (2019a) provide a theoretical explanation of the improved convergence speed of gradient clipping over standard gradient descent.

In this work, however, we reproduce the paper "Can gradient clipping mitigate label noise?" (referenced as "the paper" or "the original paper") by Menon et al. (2020) (referenced as "the authors") which focuses on *robustness* properties of gradient clipping. Informally, clipping caps the influence of any descent direction, which might help in the presence of label noise. Starting with this intuition, the authors study whether clipping can alleviate the problem of label noise studied in Ekholm and Palmgren (1982); Menon et al. (2015); Zhang and Sabuncu (2018). More specifically, they analyze the problem under symmetric label noise with the following simple linear setting: stochastic gradient descent with a linear model in a binary classification task.

Before turning our attention to the paper's experiments, which are the main focus of this reproducibility work, we state two main theoretical findings in this linear setup and the resulting novel extension of the cross-entropy loss function:

- Gradient clipping *does not* provide label noise robustness even in this simple linear setup. Specifically, clipping is linked to using a *Huberised* loss, which preserves classification-calibration but is not robust to symmetric label noise.

- A new clipping variant for composite losses is proposed, where only the contribution from the base loss is considered for clipping. The equivalent *partially Huberised* loss preserves classification-calibration and is robust to symmetric label noise.

- The resulting multi-class generalization of the partially Huberised cross-entropy loss is given in Equation 1. Suppose we have softmax probability estimates $p_\theta(x, y)$, then the *partially Huberised softmax cross-entropy loss* (PHuber-CE) is defined for $\tau > 1$ as:

$$\ell_\theta(x, y) = \begin{cases} -\tau \cdot p_\theta(x, y) + \log \tau + 1, & \text{if } p_\theta(x, y) \leq \frac{1}{\tau} \\ -\log p_\theta(x, y), & \text{otherwise.} \end{cases} \tag{1}$$

Then, the authors evaluate their partially Huberised loss in experiments on synthetic data (referenced as "synthetic experiments") to demonstrate its behavior under symmetric label noise. They show symmetric label noise scenarios that defeat the logistic loss and the Huberised logistic loss, but not the partially Huberised logistic loss. Moreover, they assess the effectiveness of partial Huberisation on real-world datasets subject to symmetric label noise (referenced as "real-world experiments"). They empirically verify that partially Huberised versions of existing losses behave well in the presence of symmetric label noise, through deep-learning experiments on the MNIST, CIFAR-10 and CIFAR-100 datasets.

We thoroughly reproduce the synthetic and real-world experiments in section 3 and section 4 respectively. Then, we evaluate the experimental results in section 5 and conclude with the assessment of empirical claims in section 6.

# 2 Background

**Gradient clipping.** Consider a supervised learning task with samples $(x, y) \in (\mathcal{X} \times \mathcal{Y}) \sim D$, and a loss function $l_\theta : \mathcal{X} \times \mathcal{Y} \to \mathbb{R}$. For this setting the gradient $g(\theta)$ and the clipped gradient $\bar{g}_\tau(\theta)$ are defined as follows:

$$g(\theta) = \frac{1}{N} \sum_{n=1}^{N} \nabla l_\theta(x_n, y_n) \qquad\qquad \bar{g}_\tau(\theta) = \begin{cases} \tau \frac{g(\theta)}{\|g(\theta)\|_2} & \text{if } \|g(\theta)\|_2 \geq \tau \\ g(\theta) & \text{otherwise.} \end{cases}$$

**Label noise.** In classification under label noise, one has samples from a noisy distribution $P_{\bar{D}}(x, y)$ instead of a clean distribution $P_D(x, y)$. For example, under *symmetric* label noise, all instances have a constant probability of their labels being flipped uniformly to any of the other classes. The task remains to minimize risk over the clean distribution $D$. Some recent loss-based proposals for learning under symmetric label noise are the linear or unhinged loss (van Rooyen et al., 2015) and the generalized cross-entropy loss (Zhang and Sabuncu, 2018).

**Huberised losses.** Huberised and partially Huberised losses, as defined in the paper, are closely related to the Huber loss (Huber, 1964), which is widely employed in robust regression. In the binary classification setting, for a predictor

$f : \mathcal{X} \to \mathbb{R}$ and labels $y \in \{\pm 1\}$, these losses are derived from the logistic loss $\phi(f(x) \cdot y) = \phi(z) = \log(1 + e^{-z})$, which can also be written as $\phi(z) = \varphi(F(z))$, with the base loss $\varphi(u) = -\log u$ and the link function $F(z) = \sigma(z)$. The *Huberised logistic loss* $\bar{\phi}_\tau$ (Equation 2) linearises the *entire* logistic loss beyond a certain threshold, while the *partially Huberised logistic loss* $\tilde{\phi}_\tau$ (Equation 3) linearises *only* the base loss but leaves the link function intact.

$$\bar{\phi}_\tau(z) = \begin{cases} -\tau \cdot z - \log(1 - \tau) - \tau \cdot \sigma^{-1}(\tau) & \text{if } z \le -\sigma^{-1}(\tau) \\ \log\left(1 + e^{-z}\right) & \text{otherwise.} \end{cases} \tag{2}$$

$$\tilde{\phi}_\tau(z) = \begin{cases} -\tau \cdot \sigma(z) + \log \tau + 1 & \text{if } z \le \sigma^{-1}\left(\frac{1}{\tau}\right) \\ \log\left(1 + e^{-z}\right) & \text{otherwise.} \end{cases} \tag{3}$$

The *partially Huberised softmax cross-entropy loss* (Equation 1) is obtained by applying that same partial Huberisation to the softmax cross-entropy loss, in which the link function is a softmax instead of a sigmoid. For more information on Huberised losses, we kindly refer to the original paper (Menon et al., 2020).

## 3 Synthetic experiments

We now study two synthetic experiments proposed by the authors to show the existence of label noise scenarios that defeat Huberised but not partially Huberised losses. We will start by discussing the 2D setting proposed in Long and Servedio (2010) and then discuss the 1D outliers setting given in Ding (2013). These experiments are fully re-implemented with NumPy (Harris et al., 2020) and SciPy (Virtanen et al., 2020). Experimental setups including methods and hyperparameters are fully verified according to the original paper and in necessary cases, according to the additional details obtained from the authors. Our experiments are configurable through the Hydra framework (Yadan, 2019). Our code re-implementing both the synthetic and real-world experiments is available at: https://github.com/dmizr/phuber

### 3.1 Long and Servedio dataset

Long and Servedio (2010) consider a set of four positive labeled points: one *large margin* example $(1, 0)$, one *puller* example $(\gamma, 5\gamma)$ and two *penalizer* examples $(\gamma, -\gamma)$ where $0 < \gamma < \frac{1}{6}$, in a binary classification task with a linear model without a bias term. The halfspace $x_1 > 0$ correctly classifies all the samples. However, one can show that under symmetric label noise, minimizing over a wide range of convex losses with a suitable $\gamma$ will result in a predictor equivalent to a random predictor.

The authors build on Long and Servedio (2010), and consider a mixture of six isotropic Gaussians $\mathcal{N}(\boldsymbol{\mu_i}, \sigma^2 I_2)$, with $\sigma = 0.01$ and $\boldsymbol{\mu_i} \in \{\pm(1, 0), \pm(\gamma, 5\gamma), \pm(\gamma, -\gamma)\} \subset \mathbb{R}^2$, with $\gamma = \frac{1}{24}$. Mixing weights are $\frac{1}{4}$ for the two Gaussians centered around $\pm(\gamma, -\gamma)$ and $\frac{1}{8}$ for the rest. An instance $(x_1, x_2)$ is labeled positive if $x_1 \ge 0$ and negative otherwise. $N = 1000$ random samples are drawn from this distribution, and the label of each sample is flipped with corruption probability $\rho < 0.5$. Then, a linear classifier is trained using Scipy's SLSQP (Sequential Least Squares Programming) optimizer for a maximum of 100 iterations with each of the following losses:

- the logistic loss
- the Huberised version of the logistic loss, with $\tau = \sigma(-1) \approx 0.26$
- the partially Huberised version of the logistic loss, with $\tau = 1 + e^{-1} \approx 1.36$

After contacting the authors, we found that the above $\tau$ values were used instead of the values provided in the original paper, which were $\tau = 1.0$ and $\tau = 2.0$ for the Huberised and the partially Huberised loss respectively.

Once trained, the classifier is evaluated on 500 clean test samples.

Figure 1a and Figure 1b show our results over 500 independent runs for $\rho = 0.45$ and $\rho = 0.2$ respectively. When using $\rho = 0.45$, as stated in the original paper, we fail to reproduce a figure that *exactly* matches the authors' results. However, through experimentation, we found that for a lower level of noise corruption such as $\rho = 0.2$, we get results that are very similar to the original paper, with the partially Huberised loss always achieving perfect classification, while the logistic and Huberised losses succumb to label noise and perform no better than chance.

### 3.2 Outliers dataset

The 1D setting from Ding (2013) is composed of 10,000 linearly separable inliers: 5000 samples from the unit variance Gaussian $\mathcal{N}(1, 1)$ with positive label, and 5000 samples from the mirror image $\mathcal{N}(-1, 1)$ with negative label. In

addition, 50 outliers are added: 25 samples from $\mathcal{N}(-200, 1)$ with positive label, and 25 samples from $\mathcal{N}(200, 1)$ with negative label. Assuming a linear model characterized by a scalar $\theta \in \mathbb{R}$, we comparatively evaluate the empirical risk with and without outliers. We use the same three losses as in subsection 3.1 but with $\tau = 0.1$ and $\tau = 1.1$ for the Huberised and partially Huberised loss respectively. [1]

Figure 1c shows our results where dashed and solid curves represent the cases with and without outliers respectively. As in the original paper, the optimal solutions for the logistic and Huberised loss are changed from $\theta^* = +\infty$ to $\theta^* = 0$ with the introduction of outliers, whereas the partially Huberised loss remains intact.

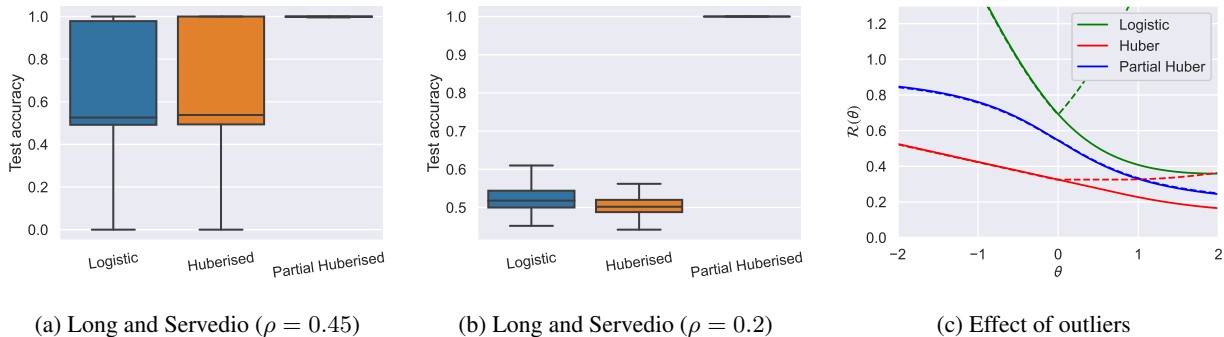

(a) Long and Servedio ($\rho = 0.45$)  (b) Long and Servedio ($\rho = 0.2$)  (c) Effect of outliers

Figure 1: Reproduction of Long and Servedio (2010) and Ding (2013) experiments. In the Ding experiment (c), the solid curve denotes empirical risk without outliers, while the dashed curve denotes empirical risk with outliers.

## 4 Real-world experiments

We now consider the deep learning experiments performed on three image classification datasets: MNIST, CIFAR-10 and CIFAR-100. These experiments were fully re-implemented with PyTorch (Paszke et al., 2019), according to the description from the paper and implementation details obtained from the authors after contacting them. Configuration management for these experiments was done with the help of the Hydra framework (Yadan, 2019).

### 4.1 MNIST

#### 4.1.1 Methodology

MNIST is a dataset of handwritten digits, consisting of a training set of 60,000 examples and a test set of 10,000 examples. Each example is a 28x28 grayscale image associated with a label from 10 distinct classes. The dataset is normalized using the mean and standard deviation from the training set, and no data augmentation is applied. The training labels are then corrupted with symmetric noise at flip probability $\rho \in \{0.0, 0.2, 0.4, 0.6\}$. As in the original paper, the same random seed is used to corrupt the training labels across all trials.

We use a LeNet-5 (Lecun et al., 1998), with a few modifications in order to reproduce the authors' settings as accurately as possible. Most notably, the $\texttt{tanh}$ activation layers from the original LeNet are changed to $ReLU$, and the weights are initialized according to a truncated normal distribution with standard deviation $\sigma = 0.1$.

This model is trained for 20 epochs using Adam (Kingma and Ba, 2017) with batch size $N = 32$, and weight decay of $10^{-3}$. The initial learning rate is set to $0.001$, and is lowered following an exponential decay schedule with decay rate $0.1$ and decay steps of $10,000$. That is, the learning rate at iteration $n$ is set to: $\eta_n = \eta_0 \cdot r^{n/s}$, with $\eta_0 = 0.001$, $r = 0.1$ and $s = 10^4$. According to the authors, these hyperparameter values were chosen to obtain a good baseline performance in a setting with no label noise.

For each level of label noise corruption, the test set accuracy of 6 different loss functions is compared:

- the cross-entropy loss (CE)
- the linear or unhinged loss (van Rooyen et al., 2015)
- the generalized cross-entropy loss (GCE), with $\alpha = 0.7$ (Zhang and Sabuncu, 2018)
- the cross-entropy loss, with global gradient clipping applied using a max norm threshold of $\tau = 0.1$

---

[1]In the original paper, the $\tau$ values mistakenly reported as 1.0 and 2.0, along with the values in subsection 3.1. These updated values are obtained from the authors, after informing them $\tau = 1.0$ for Huberisation is equivalent to keeping base loss intact.

- the partially Huberised version of the cross-entropy loss (PHuber-CE), with $\tau = 10$
- the partially Huberised version of the generalized cross-entropy loss (PHuber-GCE), with $\alpha = 0.7$ and $\tau = 10$.

The CE loss serves as a baseline, while the linear and GCE losses serve as representative noise-robust losses. The model and hyperparameters used are identical for all losses at all levels of label noise. For each of the real-world experiments and for each of the partially Huberised losses, the authors selected $\tau \in \{2, 10\}$ so as to maximize accuracy on a validation set, in a setting with flip probability $\rho = 0.6$.

### 4.1.2 Computational requirements

This LeNet model was trained with a Nvidia RTX 2080 Ti GPU. Each run took roughly 2 minutes. Fully reproducing the authors' experiments required training this model 72 times, in order to do 3 trials for each combination of loss function and level of label noise. This resulted in a total training time of around 2 hours.

### 4.1.3 Results

Our results are reported in Table 1, and a comparison with the original paper's results can be found in Figure 2. Our reproduction matches the results from the original paper for both the PHuber-CE and PHuber-GCE losses, although the CE, CE with gradient clipping and linear losses perform considerably better at high levels of label noise than what was reported. As a consequence, the partially Huberised version of these losses do not outperform the base losses at high levels of label noise, contrary to the original paper's results. It is of note that in our reproduction, all losses, except for the CE loss with gradient clipping, perform comparably, with a test accuracy higher than 97.5% at all levels of label noise.

| Dataset | Loss function | $\rho = 0.0$ | $\rho = 0.2$ | $\rho = 0.4$ | $\rho = 0.6$ |
|---------|---------------|--------------|--------------|--------------|--------------|
| MNIST | CE | 99.1 ± 0.1 | 98.8 ± 0.0 | 98.6 ± 0.0 | 98.0 ± 0.1 |
| | CE + clipping | 97.0 ± 0.0 | 96.5 ± 0.0 | 95.7 ± 0.1 | 94.7 ± 0.1 |
| | Linear | 95.0 ± 3.5 | 98.5 ± 0.1 | 98.2 ± 0.0 | 97.6 ± 0.0 |
| | GCE | 98.8 ± 0.0 | 98.7 ± 0.0 | 98.5 ± 0.0 | 98.1 ± 0.0 |
| | PHuber-CE $\tau = 10$ | 99.0 ± 0.0 | 98.8 ± 0.1 | 98.5 ± 0.1 | 97.6 ± 0.0 |
| | PHuber-GCE $\tau = 10$ | 98.9 ± 0.0 | 98.7 ± 0.0 | 98.4 ± 0.0 | 98.0 ± 0.0 |

Table 1: Reproduction of the MNIST experiments. The mean and standard error of the test accuracy over 3 trials is reported. The highlighted cells correspond to the best performing loss at a given $\rho$.

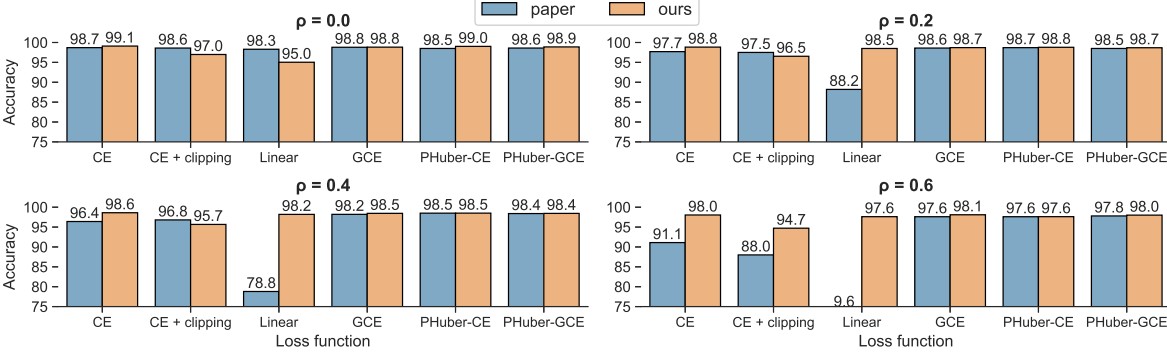

Figure 2: Test accuracy of LeNet-5 on MNIST

## 4.2 CIFAR-10 and CIFAR-100

### 4.2.1 Methodology

The CIFAR-10 and CIFAR-100 datasets (Krizhevsky and Hinton, 2009) both consist of a training set of 50,000 examples and a test set of 10,000 examples. Each example is a $32 \times 32$ color image, associated with a label from 10 distinct classes for CIFAR-10, and 100 distinct classes for CIFAR-100. Both datasets are normalized using per-channel mean and standard deviation, and the standard data augmentation for these datasets is applied, akin to Zagoruyko and Komodakis (2016). That is, images are zero-padded with 4 pixels on each side to obtain a $40 \times 40$ image, and then a random $32 \times 32$

crop is extracted and mirrored horizontally with 50% probability. As in the MNIST experiment, the training labels are corrupted with symmetric noise at flip probability $\rho \in \{0.0, 0.2, 0.4, 0.6\}$, with identical noise seed across trials.

For both of these experiments, we use a ResNet-50 (He et al., 2015), as implemented in Liu (2017). This implementation of the ResNet-50 differs from the one described by He et al. (2015) to make it more appropriate for classification on small images. The number of filters per layer is identical, but the first layer, originally a 7x7 convolutional layer with stride 2 and padding 3, is changed to a 3x3 convolutional layer with stride 1 and padding 1, and the max-pooling layer that follows is removed. By removing these early downsampling layers, this architecture performs better on CIFAR-10 and CIFAR-100 than the original ResNet-50 [2], which was designed for classification on ImageNet (Deng et al., 2009). We decided to use such an implementation for several reasons: First, it is used in many popular papers performing classification on CIFAR with ResNets, such as DeVries and Taylor (2017); Zhang et al. (2018); Li et al. (2020); Zhang et al. (2019b). Second, using the original ResNet-50 yielded poor results, especially on partially Huberised losses. Third, after contacting the authors about their implementation, they confirmed using a ResNet-50 with some of the early downsampling layers removed, but could not provide more details as to which layers were specifically changed or removed.

For CIFAR-10, this ResNet is trained for 400 epochs using SGD with Nesterov momentum 0.1 (Nesterov, 1983; Sutskever et al., 2013), batch size $N = 64$, and weight decay of $5 \times 10^{-4}$. [3] The initial learning rate is set to 0.1 and is divided by 10 at the 160th, 300th and 360th epoch. For CIFAR-100, this ResNet is trained for 200 epochs using SGD with Nesterov momentum 0.1, batch size $N = 128$, and weight decay of $5 \times 10^{-4}$. [4] The initial learning rate is set to 0.1 and is divided by 5 at the 60th, 120th and 160th epoch. According to the authors, these hyperparameters were partially based on the setting from DeVries and Taylor (2017), and were chosen to obtain a good performance with CE in a setting with no label noise.

As in the MNIST experiment, the test set accuracy of the CE, CE with gradient clipping, linear, GCE, PHuber-CE and PHuber-GCE losses are compared. The tunable parameters for these losses are identical to the ones used in the MNIST experiment, except for PHuber-CE for CIFAR-10, where $\tau = 2$. The model and hyperparameters used are identical for all losses at all levels of label noise.

We also report an additional experiment, where we train a model on CIFAR-100 using the PHuber-CE loss with $\tau = 50$. This corresponds to linearizing the base loss at probability threshold 0.02.

### 4.2.2 Computational requirements

We use a Nvidia RTX 2080 Ti GPU to train these models. With full precision training, a run on CIFAR-10 takes approximately 11 hours, while a run on CIFAR-100 takes approximately 4 hours, due to the lower amount of epochs and higher batch size. In order to accelerate the training process, we implement mixed precision training (Micikevicius et al., 2017), which results in a 2x speed-up with no decrease in accuracy compared to full precision training.

Fully reproducing the authors' experiments required training each model 72 times, resulting in a total training time of around 400 hours for the CIFAR-10 experiments, and around 150 hours for the CIFAR-100 experiments.

### 4.2.3 Results

Our results are reported in Table 2, and a comparison with the original paper's results can be found in Figure 3 and Figure 4.

On CIFAR-10, our reproduction achieves comparable or better results than the original paper for nearly all configurations, except for the Linear, GCE and PHuber-GCE losses which perform worse for $\rho = 0.6$. Surprisingly, the CE loss with gradient clipping performs considerably better than what was reported in the presence of label noise, achieving the second-highest accuracy for $\rho = 0.6$, behind PHuber-CE. Similar to the original paper's results, PHuber-CE with $\tau = 2$ is competitive with CE in the absence of label noise, and achieves very good results under label noise, outperforming all the other losses. Notably, in our reproduction, PHuber-CE outperforms the linear loss for $\rho = 0.4$, which was not the case in the original paper.

---

[2]In the ResNet paper, He et al. also propose ResNet architectures suited for CIFAR-10 classification, such as the ResNet-44 and ResNet-56, which have fewer filters per layer compared to the implementations from Liu (2017), resulting in faster training at the cost of lower accuracy. These ResNet architectures were not used in our reproduction as the authors specifically mentioned using a ResNet-50.

[3]In the original paper, the weight decay is mistakenly reported as $5 \times 10^{-3}$, and it was not specified that the type of momentum used was Nesterov momentum. These updated hyperparameters were obtained from the authors, after informing them of our difficulty reproducing their experiments with the values from the paper.

[4]See previous footnote.

On CIFAR-100, our reproduction achieves better results than the original paper for nearly all configurations. Most notably, the accuracy of the CE, GCE and PHuber-GCE losses are noticeably better at all levels of noise corruption. As in the original paper, PHuber-GCE with $\tau = 10$ achieves the best accuracy out of all losses for $\rho = 0.4$ and $\rho = 0.6$, and performs comparably to GCE for $\rho = 0.0$ and $\rho = 0.2$. Unlike the paper's results, PHuber-CE with $\tau = 10$ performs quite poorly compared to CE, even in settings with high levels of label noise where it should supposedly perform well. However, with our additional experiment using PHuber-CE with $\tau = 50$, we show that there exist values of $\tau$ for which PHuber-CE performs comparably to CE in the noise-free case, and outperforms CE at high levels of label noise.

| Dataset | Loss function | $\rho = 0.0$ | $\rho = 0.2$ | $\rho = 0.4$ | $\rho = 0.6$ |
|---------|---------------|--------------|--------------|--------------|--------------|
| CIFAR-10 | CE | 95.8 ± 0.1 | 84.0 ± 0.3 | 67.8 ± 0.3 | 44.0 ± 0.2 |
| | CE + clipping | 89.3 ± 0.0 | 82.6 ± 1.6 | 78.7 ± 0.2 | 67.6 ± 0.1 |
| | Linear | 94.1 ± 0.1 | 91.4 ± 0.5 | 86.0 ± 2.4 | 58.6 ± 5.2 |
| | GCE | 95.3 ± 0.0 | 92.5 ± 0.1 | 82.4 ± 0.1 | 53.3 ± 0.3 |
| | PHuber-CE $\tau = 2$ | 94.8 ± 0.0 | 92.8 ± 0.2 | 87.8 ± 0.2 | 73.2 ± 0.2 |
| | PHuber-GCE $\tau = 10$ | 95.4 ± 0.1 | 92.2 ± 0.2 | 81.5 ± 0.2 | 54.3 ± 0.5 |
| CIFAR-100 | CE | 75.4 ± 0.3 | 62.2 ± 0.4 | 45.8 ± 0.9 | 26.7 ± 0.1 |
| | CE + clipping | 23.5 ± 0.2 | 20.4 ± 0.4 | 16.2 ± 0.5 | 12.9 ± 0.1 |
| | Linear | 13.7 ± 0.7 | 8.2 ± 0.3 | 5.9 ± 0.7 | 3.9 ± 0.3 |
| | GCE | 73.3 ± 0.2 | 68.5 ± 0.3 | 59.5 ± 0.5 | 40.3 ± 0.4 |
| | PHuber-CE $\tau = 10$ | 60.6 ± 1.1 | 54.8 ± 1.2 | 43.1 ± 1.1 | 24.3 ± 0.8 |
| | PHuber-GCE $\tau = 10$ | 72.7 ± 0.1 | 68.4 ± 0.1 | 60.2 ± 0.2 | 42.2 ± 0.4 |
| | PHuber-CE $\tau = 50$ | 75.4 ± 0.2 | 65.9 ± 0.2 | 49.1 ± 0.2 | 26.9 ± 0.0 |

Table 2: Reproduction of the CIFAR-10 and CIFAR-100 experiments. The mean and standard error of the test accuracy over 3 trials is reported. The highlighted cells correspond to the best performing loss at a given $\rho$. CIFAR-100 PHuber-CE with $\tau = 50$ is an additional experiment that was not performed in the original paper.

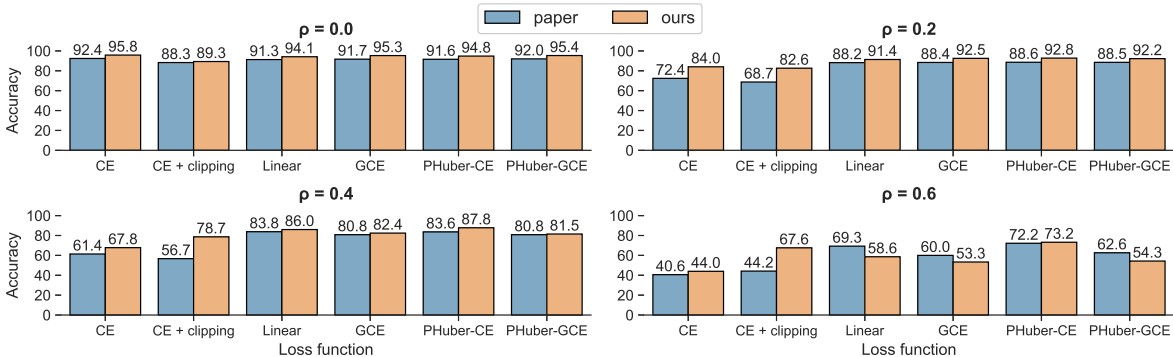

Figure 3: Test accuracy of ResNet-50 on CIFAR-10

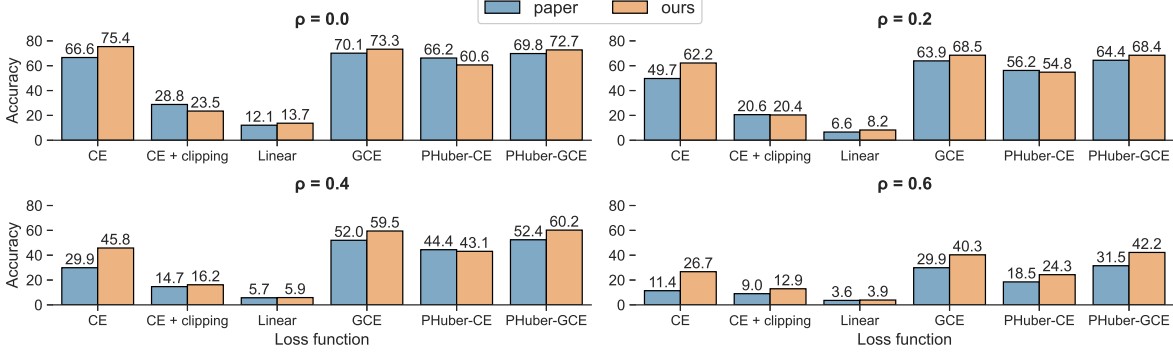

Figure 4: Test accuracy of ResNet-50 on CIFAR-100

# 5 Discussion

We now discuss whether our experimental results support the claims of the paper.

For the Long and Servedio experiment, when reusing exactly the parameters described in the paper and in the authors' clarifications, our results do not perfectly match those from the original paper. After experimenting with some of the parameters, we did find values that produce results nearly identical to those shown in the paper. Nonetheless, these results still support the claim the authors made for the synthetic experiments, namely, that there exist label noise scenarios for both the Long and Servedio (2010) setting and the Ding (2013) setting which defeat a Huberised but not a partially Huberised loss. While we made a considerable effort to ensure that our implementation matches the paper's description, the difference in results could be due to some minor differences in our implementations, or to a difference in the random seeds used to sample from the mixture model and flip the labels.

For the MNIST experiment, all losses, except for CE with clipping, perform comparably, achieving very high accuracy for all levels of label noise. This differs from the results of the original paper, where CE and linear losses were affected by label noise. As a result, it is difficult to support or reject any claim made regarding these losses with this experiment.

For both the CIFAR-10 and CIFAR-100 experiments, our results differ even after fixing the hyperparameters which were accidentally misreported in the original paper, with our implementation yielding a noticeably higher test accuracy for the CE loss with clipping on CIFAR-10, and the CE, GCE and PHuber-GCE losses on CIFAR-100. This is likely due to the ResNet-50 architecture used, as the generally higher accuracy could be explained if our model happens to have a higher number of parameters than theirs. The deep learning framework used could also lead to different results, as the authors mentioned using TensorFlow while we used PyTorch. Finally, this could also be caused by the random seed used to add label noise, although we did not notice any significant difference in results when changing this seed.

For the CIFAR-10 experiment, our reproduction supports the claim that for these specific hyperparameters, partially Huberised losses are competitive with the base loss in the noise-free case and can outperform it under label noise. In addition, this experiment also shows that PHuber-CE can be very effective at mitigating symmetric label noise, as it performs considerably better than the representative noise-robust losses at high levels of label noise.

For the CIFAR-100 experiment, our reproduction shows that PHuber-GCE loss with $\tau = 10$ is competitive with the base loss (GCE) in the noise-free case and can outperform it at high levels of label noise, which supports the aforementioned claim. However, this claim does not hold for the PHuber-CE loss with $\tau = 10$, which performs worse than CE in all cases. Despite that, we show with our additional experiment that there exists a value of $\tau$ for which the PHuber-CE loss performs comparably to CE in the noise-free case, and improves upon it under label noise.

Our additional experiment shows that the value of $\tau$ plays a crucial role in the performance of partially Huberised losses. Both the PHuber-CE and GCE losses interpolate between the linear and the CE loss. PHuber-CE and GCE mimic the linear loss for $\tau \to 1$ and $\alpha = 1$ respectively, while for $\tau \to +\infty$ and $\alpha \to 0$, they mimic the CE loss. As the linear loss fails to train properly in our CIFAR-100 experiment, it is expected to obtain poor results for these losses if the tunable parameter used makes them too similar to the linear loss. As PHuber-GCE combines both of these losses, it can also perform poorly in such a scenario. Furthermore, the CE loss with gradient clipping also has a tunable parameter which strongly affects performance, as our reproduction shows that for a max norm $\tau = 0.1$, CE with clipping can perform significantly better than CE on CIFAR-10, and significantly worse on CIFAR-100.

In order to properly compare these losses, it would therefore be of interest to find, for each level of label noise, the tunable parameter values for which they perform best, by using random search or a hyperparameter tuning framework such as Optuna (Akiba et al., 2019) on a validation set. While such a hyper-parameter search has a high computational cost, it would offer some valuable insights on how well each of these losses performs, and how sensitive they are to changes to their tunable parameters. We leave such exploration for future work.

# 6 Conclusion

In this work, we fully re-implement the experiments performed in Menon et al. (2020). For the synthetic experiments, our results differ when using the exact values described in the paper, although they still support the main claim, and by slightly modifying some experiment settings, we obtain results almost identical to those of the original paper. Our results also differ for the deep learning experiments, with some of the baselines performing better than described. Nonetheless, these experiments still support the claim that partially Huberised losses perform well on real-world datasets subject to label noise. Our additional experiment also provides further insight on the performance of partially Huberised losses, as it empirically shows that the value of $\tau$ can play an important role in the performance of models trained with these losses. We thus believe it would be of interest to perform further experiments focused on tuning these losses for different levels of label noise, although this would incur a relatively high computational cost.

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

# A  Decision boundaries in the Long and Servedio experiment

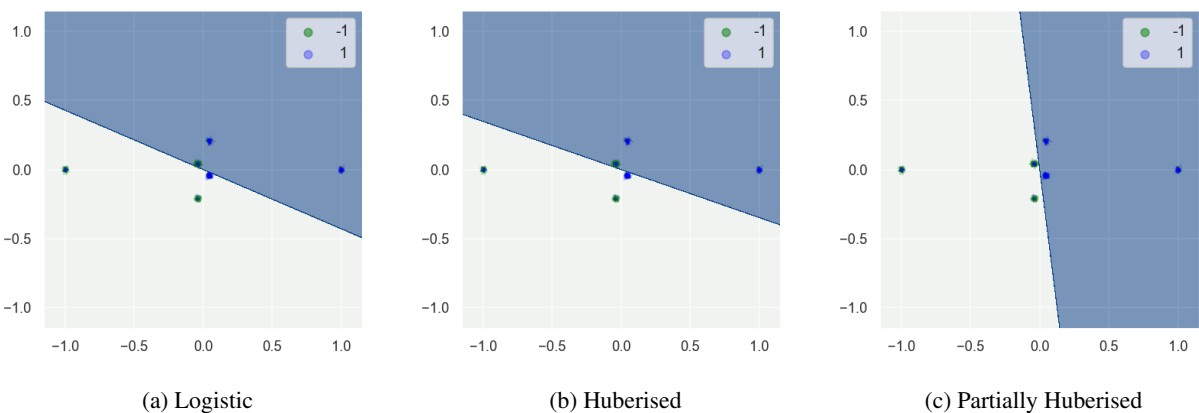

| (a) Logistic | (b) Huberised | (c) Partially Huberised |
|---|---|---|

Figure 5: Decision boundaries in the reproduced Long and Servedio (2010) experiment, with label noise flip probability $\rho = 0.2$. The logistic and Huberised logistic loss misclassify samples generated from the two *penalizer* Gaussians centered around $\pm(\gamma, -\gamma)$, resulting in $50\%$ test accuracy. The partially Huberised logistic loss does not succumb to label noise, achieving near-perfect discrimination.

# B  Effect of the label noise seed on the real-world experiments

In the original paper (Menon et al., 2020), all trials use the same corrupted dataset for each flip probability $\rho$. Table 3 shows the results obtained on a subset of our experiments when varying the random seed used to generate label noise.

| Label noise | Dataset | Loss function | Seed 0 | Seed 1 | Seed 2 | Seed 3 | Original paper |
|---|---|---|---|---|---|---|---|
| | MNIST | CE | 98.0 ± 0.1 | 97.9 ± 0.0 | 97.9 ± 0.0 | 97.9 ± 0.1 | 91.1 ± 0.6 |
| | | PHuber-GCE | 98.0 ± 0.0 | 97.8 ± 0.1 | 98.0 ± 0.0 | 98.0 ± 0.0 | 97.8 ± 0.0 |
| $\rho = 0.6$ | CIFAR-10 | CE | 44.0 ± 0.2 | 43.8 ± 0.5 | 44.1 ± 0.2 | 43.2 ± 0.3 | 40.6 ± 0.3 |
| | | PHuber-GCE | 54.3 ± 0.5 | 54.0 ± 0.7 | 53.7 ± 0.5 | 54.3 ± 0.2 | 62.6 ± 0.2 |
| | CIFAR-100 | CE | 26.7 ± 0.1 | 27.1 ± 0.1 | 27.0 ± 0.5 | 26.9 ± 0.1 | 11.4 ± 0.2 |
| | | PHuber-GCE | 42.2 ± 0.4 | 43.0 ± 0.2 | 42.1 ± 0.6 | 41.9 ± 0.1 | 31.5 ± 0.8 |

Table 3: Impact of the random seed used to generate label noise. The mean and standard error of the test accuracy over 3 trials is reported. This subset of experiments was chosen to include both a baseline and a partially Huberised loss, at the highest level of label noise. The results obtained are consistent across seeds, and differ from the original paper's results.

