# OpenReview forum: "[Re] Can gradient clipping mitigate label noise?"
_ML_Reproducibility_Challenge/2020 — RC2020_

### Official Review · AnonReviewer1 · 2021-03-01
**Solid report and solid reproductions**

**Rating:** 8
**Confidence:** 4

**Review:**

In this report, the authors reproduce the experiment parts of the paper: "Can Gradient Clipping Mitigate Label Noise?" by Aditya Krishna Menon, Ankit Singh Rawat, Sashank J. Reddi, Sanjiv Kumar.

The main contributions of this report are:
1. The authors reproduce experiments of the original paper to prove that Partially Huberised Loss does have label noise robustness.
2. The authors report corrected hyperparameters that were reported incorrectly in the original paper.
3. Although it's not so much different from what the original paper insists, the authors provide results that are different from the original paper.
4. The authors clarify that the ResNet-50 architecture used in the original paper differs from the architecture of He et al. 2015 (this choice was not made clear in the original paper) and explains why this modification is necessary.

Minor issue:
In experiments using Long & Servedio dataset, there is no explanation for why is the accuracy becomes lower when the corruption rate ρ is lower.


Verdict:
The report contains some solid experiments and additional information and corrections not in the original paper. I can see researchers trying to build upon the original paper benefitting from reading this reproducibility report. For this reason, I recommend the report be accepted.



**Familiar With The Original Paper:**

I have read the original paper

**Reproducibility Summary:**

Report has summary

---

### Official Review · AnonReviewer3 · 2021-03-01
**Good reproducability report**

**Rating:** 7
**Confidence:** 4

**Review:**

The report aims to reproduce a paper on so-called Partially Huberised losses. These losses are used to mitigate the label noise. The report replicates all the experiments from the original paper and makes insightful discoveries.

The report is well-written and easy to follow. The original paper is summarized and all the experiments are described in details. The report also mentions the communication with the authors of the original paper which revealed some typos or mistakes in the original paper.

The original paper was reproduced from scratch with a different framework (PyTorch, instead of Tensorflow). This is good way to test the reproducability.

Suggestions for improvements:
- The very first section "Scope of Reproducability" should discuss, the scope. What the report reproduces and what it does not.
- The hyper-parameter search is the same as in the original paper. The report does not go an extra mile to search wider space. Nevertheless, I understand that such experiments may be time consuming and require a lot of computational resources.
- It is always desirable to include extra datasets, extra experiments, more architectures to more thoroughly test the claims of the original paper.
- Section 4.2.2 mentions that the reproduced experiments used the mixed precision for training. The report should clarify if this is different from the original paper and if it may influence the results.

Overall, this is a good report. I recommend accept.

**Familiar With The Original Paper:**

I have read the original paper

**Reproducibility Summary:**

Report has summary

---

### Official Review · AnonReviewer2 · 2021-03-08
**Review for Can gradient clipping mitigate label noise?**

**Rating:** 7
**Confidence:** 3

**Review:**

The report aims to reproduce the results of the paper 'Can gradient clipping mitigate label noise?' The report gives a summary of how the reproduction is conducted and briefly introduce the original paper. The paper also gives the details including the hyper-parameters and computational infrastructures.

The authors provides the reproducing codes with detailed documentation.

**Familiar With The Original Paper:**

I have not read the original paper

**Reproducibility Summary:**

Report has summary

---

### Decision · Program_Chairs · 2021-03-31

**Decision:**

Accept

**Comment:**

Super complete, well-presented and explained report, with a strong discussion/conclusion